# The Importance of Habitat and Lake Morphometry for the Summer Diet Choice of Landlocked Arctic Char in Two West Greenland Lakes

Andreas S. Berthelsen [1,2,*], Katrine Raundrup [3], Peter Grønkjær [4,5], Erik Jeppesen [1,2,5,6,7] and Torben L. Lauridsen [1,2,5]

1   Department of Ecoscience—Lake Ecology, Aarhus University, C.F. Møllers Alle Building 1131, 8000 Aarhus, Denmark; ej@ecos.au.dk (E.J.); tll@ecos.au.dk (T.L.L.)
2   Sino-Danish Centre for Education and Research, Beijing 100049, China
3   Greenland Institute of Natural Resources, Postbox 570, Nuuk 3900, Greenland; kara@natur.gl
4   Department of Biology—Aquatic Biology, Aarhus University, Ole Worms Allé 1, 8000 Aarhus, Denmark; peter.groenkjaer@bio.au.dk
5   Arctic Research Centre, Aarhus University, Ole Worms Allé 1, 8000 Aarhus, Denmark
6   Limnology Laboratory, Department of Biological Sciences and Centre for Ecosystem Research and Implementation, Middle East Technical University, Ankara 06800, Turkey
7   Institute of Marine Sciences, Middle East Technical University, Mersin 33731, Turkey
*   Correspondence: asbert@ecos.au.dk; Tel.: +45-2495-4504

**Abstract:** Arctic char (*Salvelinus alpinus*) is a top predator and the most widespread fish in Arctic lakes. The presence of Arctic char affects the predator–prey dynamics of the key species in the food webs in these lakes. This study sought to elucidate the effects of habitat (littoral, pelagic, or profundal) and lake morphometry on the trophic position of this char in the food web. Using stomach content and stable isotope analyses, we investigated the effect of fish length, habitat, and time (individual survey years: 2008, 2013, 2018, and 2019) on the dietary niches of landlocked Arctic char populations during summer in two west Greenland lakes: Badesø (area 0.8 km$^2$, mean depth 9.2 m) and Langesø (area 0.3 km$^2$, mean depth 5.0 m). The small char (<20 cm fork length) in Badesø generally foraged less littoral macroinvertebrates than those from Langesø. The large chars were mainly piscivorous in both lakes. In Badesø, there was a shift from relying on littoral to pelagic invertebrates by the small char from 2008–2013 to 2018–2019. The proportionally larger size of the littoral habitat in the smaller Langesø led to an increased reliance on littoral-derived macroinvertebrates in the diet of the small char, more so than in the larger Badesø, where the predominant reliance was on pelagic sources.

**Keywords:** trophic position; piscivory; zooplanktivory; lake size; fish length

## 1. Introduction

Freshwaters contain almost half of the world's fish species, yet only a fraction of aquatic habitats [1], and only few of these species occur in Arctic lakes. The trophic structure, habitat selection, and feeding of fish often depends on the nutrient level, lake size, and habitat heterogeneity [2–6], as well as the fish community composition and interactions [2,7,8]. Disentangling these dynamics is difficult in lakes with many fish species. Due to their simplified food webs, the fish communities in Arctic lakes are, however, ideal for studying the trophic dynamics and food selection in these lakes. Arctic char (*Salvelinus alpinus*) is the most widespread freshwater fish in Greenland [9] and is a valued species both commercially and culturally [10]. Furthermore, Arctic char play a fundamental role in the trophic coupling between the pelagic and benthic pathways in such lakes [7,11,12]. Char populations vary substantially in both their morphology and ecology, depending on the environment [13,14]. Several previous studies have examined the dietary niche of char, but

have mainly focused on the role of char morphs/ecotypes [11,14–18] or on the diet of char compared to other fish species [19–21].

Arctic char is the primary top predator in Greenlandic lakes [8]. In the absence of specialized ecotypes, it preferentially feeds on the most energetically viable food items, selecting these food items based on prey size and accessibility [22]. The most common prey groups for char in west Greenland are midges (chironomids), small-bodied crustacean zooplankton (*Bosmina* spp. and *Holopedium gibberum*), and, if present, three-spined sticklebacks (*Gasterosteus aculeatus*; [8,23,24]).

Lake morphometry plays an important role in the dietary niche of Arctic char [15,19,20]. Furthermore, lake size has been found to be important in determining the arctic char size and community structure in Greenlandic lakes [15]. The relative size of habitats is essential in structuring lake food webs and for determining the trophic role of char, due to habitat-specific differences in prey abundance [20]. The prey items in Greenlandic lakes vary as to their preferred habitat. Zooplankton are more abundant in the pelagic habitat [25], while invertebrate zoobenthos and sticklebacks are most abundant in the littoral habitat [26]. In accordance with the square-cube law, the relative volume of the pelagic habitat in deep lakes increases disproportionally with area. Accordingly, the relative importance of the pelagic habitat as a foraging ground also increases disproportionally with lake size. The production per unit area in clear, nutrient-poor lakes is often highest in the littoral habitat [27]; therefore, relative primary production is high when the littoral area is proportionally larger compared to the pelagic and profundal habitats [27,28].

Arctic chars prefer to forage in the littoral habitat [29]. Based on metabolic costs, it is reasonable to assume that this is caused by a "constant satiation" foraging strategy [22]. However, when their resources are scarce, chars have also been found to feed on pelagic food items [12]. Knowledge of the trophic position of char is therefore key in determining the respective constraints that this trophic position of char enforces on the food web, community structure, food web robustness, and the potential for trophic cascades.

Stomach content and stable isotope analyses are proven methods for investigating the dietary niche and trophic position of fish [30]. An examination of the identifiable prey from stomach contents is the most direct method of providing a short-term snapshot of feeding (hours or days). However, for this method to be used effectively, factors such as the digestibility of these diet items should be taken into account. Furthermore, a stomach content analysis generally requires a large sample size to capture the full range of prey items.

A stable isotope analysis (e.g., $\delta^{15}N$ and $\delta^{13}C$) can be used to make a more long-term assessment of diet and the trophic position (months). However, the use of stable isotopes in fish diet analyses requires definite knowledge of prey sources, in order to avoid leaving out important diet items. Furthermore, because of the relatively slow turnover of the stable isotopes in fish tissue, short-term changes in the diet may not be discovered by a stable isotopes analysis alone [31]. Using a combined stable isotope and stomach content analysis strengthens both methods [32]. In this study, we used both stable isotope and stomach content analysis.

We investigated the effects of habitat and lake morphometry on the diet and trophic position of chars in two interconnected Greenlandic lakes, varying in size and depth, and at a temporal scale covering 11 years (2008–2019). Both lakes hosted three-spined sticklebacks. We hypothesized that the diet composition of the char would be more habitat-dependent in the larger and deeper Badesø lake than in the smaller and shallower Langesø lake. Furthermore, based on previous studies, we expected the size of the individual fish to have a significant effect on their trophic position and diet choice [15,33].

## 2. Materials and Methods

Badesø and Langesø are located in the Kobbefjord catchment area (64°07′50″ N, 51°21′38″ W), ca. 20 km from Nuuk, Greenland. These two nutrient-poor lakes are connected and thus partly share the same catchment. They are situated close to each other geographically (Figure 1). These two lakes were chosen because both have a population of

landlocked char and, although interconnected, they differ in their morphometry. Badesø is larger, deeper, and less elongated than Langesø (Table 1), and the relative volume of the pelagic habitat is therefore larger in Badesø, while the relative size of the littoral habitat is larger in Langesø.

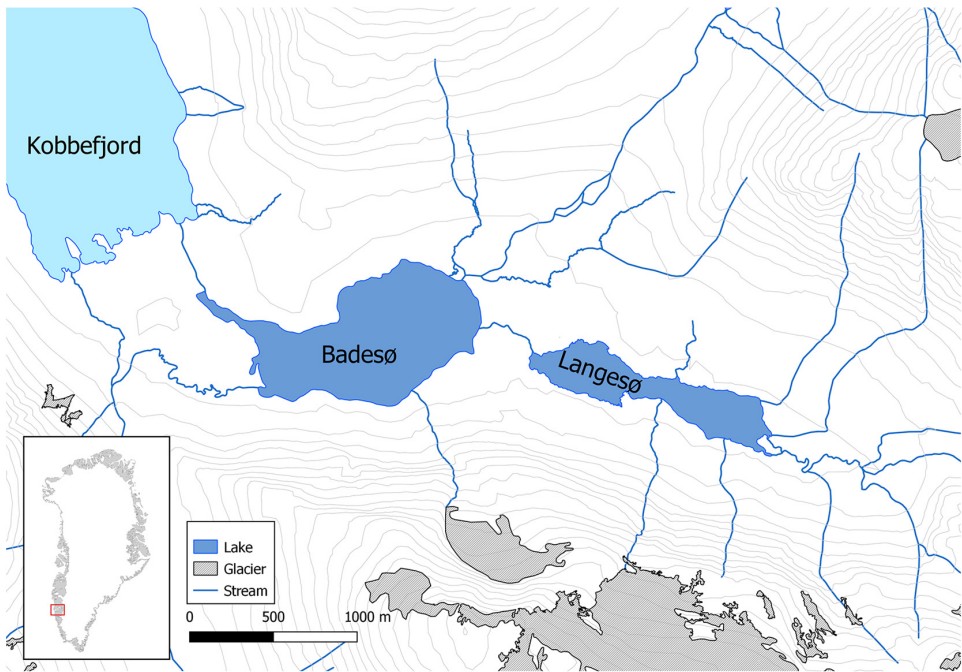

**Figure 1.** Map of the study lakes, Badesø and Langesø, in Greenland.

**Table 1.** Bathymetric and chemical characteristics of the two study lakes, Badesø and Langesø. Badesø is based on yearly averages from 2008–2020. Langesø is based on a single measurement from 2019.

|  | **Badesø** | **Langesø** |
|---|---|---|
| Surface area (ha) | 80 | 30 |
| Max depth (m) | 38 | 16 |
| Average depth (m) | 9.2 | 5 |
| Elevation (m above sea) | 35 | 40 |
| Total phosphorus (mg/L) | 0.006 | 0.007 |
| Total nitrogen (mg/L) | 0.078 | 0.038 |

The Arctic char and three-spined stickleback were sampled in 2008 (20–3/8), 2013 (6–7/8), 2018 (31/7–02/8), and 2019 (11–15/7) in Badesø. Those from Langesø were sampled in 2013 (7–13/8) and 2019 (11–15/7). In both lakes, we used sinking Lundgren multi-mesh monofilament nylon gillnets that were 1.5 m high and 42 m long, with 14 mesh sizes from 6.25 to 75 mm. Three nets were set in each of the three habitats: littoral, pelagic, and profundal. The littoral nets were placed at the bottom, at a depth of 2–2.5 m. The pelagic nets were placed at a water depth of 15–20 m and the nets were suspended 6–8 m below the surface. The profundal nets were placed near the sediment in areas with a water depth of 18–20 m in Badesø and at a 10–11 m depth in Langesø. All the nets were set parallel to the shore late in the afternoon and left for approximately 16 h. The fork length and fresh weight were measured for all the fish caught. The char stomachs (from pharynx to pylorus) were stored individually in 96% ethanol. The samples of muscle tissue for the isotope analysis were taken from the dorsal muscles, kept in closed containers, and frozen. The intact three-spined sticklebacks were sealed in containers and frozen. The dorsal muscle tissue from these was later processed for the isotope analyses.

Macroinvertebrates were sampled from the littoral habitat using a sweep net (mesh size: 500 μm). The only macroinvertebrates found in a sufficient quantity to be included

in the isotope analyses were chironomid larvae. Two differently sized categories of zooplankton were collected from the pelagic habitat with plankton nets (mesh size 140 μm and 500 μm). Only one zooplankton category (mesh size 140 μm) was included during the Badesø surveys in 2008, 2013, and 2018. After collection, the samples were kept frozen until they could be freeze-dried and processed for the stable isotope analysis.

We analyzed the fish catch per unit effort (CPUE) and fork length data, and both were log transformed to improve the normality. We tested for differences between the habitats, years, and lakes using a combination of ANOVA and Tukey HSD tests.

All the sampled stomachs ($n$ = 395) from the years 2008, 2013, 2018, and 2019 were analyzed using the points method, as described by Hynes 1950 [34]. First, the stomach fullness was determined visually for each stomach using points from 0 (empty) to 5 (full). Then, the diet items were assigned points from 1 to 10 based on their relative contribution to the stomach fullness. The points gained by each food item were summed and scaled down to percentages for each stomach. All the non-empty stomachs were always assessed using all 10 points.

The char stomach samples were divided into groups based on fish length (small: <20 cm or large: >20 cm fork length), as well as the habitat (littoral, pelagic, and profundal/benthic) in which the fish were caught. The length groups were based on the degree of piscivory (Figure 2). Fish larger than 20 cm had a high probability of piscivory. There was no significant difference in the length at which the char turned to piscivory between the lakes ($X^2_{df = 1.342}$ = 3.48, $p$ = 0.062). The relative contribution of the different prey items to the diets of these groups (length and habitat) was compared by calculating a square root transformed Bray–Curtis similarity matrix. The factors (fish length, year, and habitat) were then analyzed using an ANOSIM (analysis of similarities). ANOSIM is a non-parametric ANOVA-like test [35] calculated using Primer 7 software (version 7.0.21, Albany, New Zealand). A square root transformation was chosen to reduce the statistical impact of the common item groups, such as three-spined sticklebacks and chironomid larvae, for the ANOSIM tests. The stomach content sample groups were also analyzed using a principal component analysis (PCA), using a modified version of Crespin De Billy et al. (2000) [36], which based this on fish groups rather than individuals. For the PCA, the chars were grouped according to their habitat, year, and size.

The muscle tissue samples from the char were analyzed, using the fish length (<20 cm or >20 cm fork length), habitat (littoral, pelagic, and profundal/benthic), and year in which the fish were caught. For the stable carbon and nitrogen isotope analysis, 20 fish per size group and habitat were selected randomly, thus representing the full length range per group and habitat. If a group contained less than 20 fish, they were all included. All the isotope samples of the potential food items were collected in the same period as the fish sampling. For macroinvertebrates and zooplankton, three replicates per sample type were analyzed, except for Badesø in 2008 and 2013, where only one sample was available.

The stable isotope samples were freeze dried for 24 h, homogenized by being ground into a fine powder, weighed (0.5–2 mg tissue), and packed into tin capsules for the stable isotope analysis (SIA) at the stable isotope lab at the Center for Geomicrobiology in Aarhus, Denmark. Values of $\delta^{15}$N and $\delta^{13}$C were determined in an elemental analyzer using isotope ratio mass spectrometry (EA-IRMS). The results are presented in the usual δ notation, relative to the atmospheric N2 gas for nitrogen and atmospheric CO2 for carbon. The carrier gas for both was helium. The stable isotope data are expressed in parts per thousand (‰).

In 2008 and 2013, USGS41 and N2AIR [37] were used as reference materials, together with the secondary isotopic reference materials glutamic acid, bovine liver, and nylon 5. The international standards USGS40 [37], NBS18 [38], and N2AIR were used as reference materials in 2018 and 2019.

To elucidate the food structures for the large and small char, we used Bayesian inferences in the simmr R package [39]. Because of the relatively high observed C:N ratio in the char muscles, the char muscle samples were lipid-normalized using the method described in Kiljunen et al. (2006) [40]. We did not lipid correct the prey organisms, as the removal of

these lipids would contradict the assumption of the isotopic mass balance and complete mixing during a trophic transfer [41,42]. The prey items were zooplankton from the pelagic zone, chironomid larvae and pupae from the littoral zone, and sticklebacks representing the primary source of piscivory. Only the prey items commonly found in the chars' stomach contents were included in the analysis. We used the commonly applied trophic enrichment factors $0.4 \pm 1.3‰$ for $\delta^{13}C$ and $3.4 \pm 1.0‰$ for $\delta^{15}N$, as suggested by Post (2002) [43] and used in previous char studies by Eloranta et al. (2013) [12]. The data used in the models are shown as isospace plots, while the results of the simmr models are shown as ternary plots.

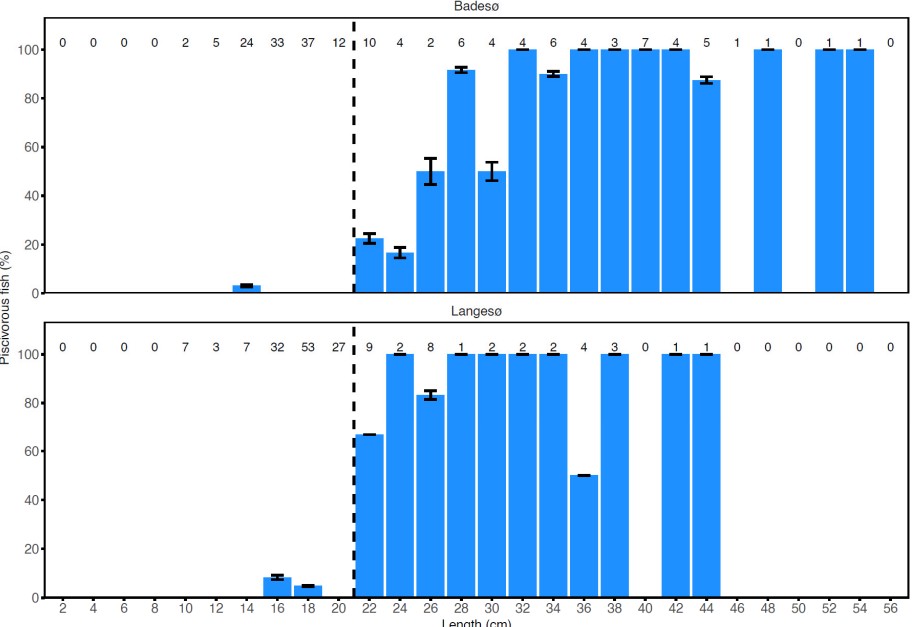

**Figure 2.** Average percent of piscivorous char from different lengths in Badesø and Langesø. Dashed line indicates where the cutoff for fish length groups were assigned. Standard error between the study years indicated by error bars. Number of fish in each group indicated above each column. Empty stomachs were not included in the figure.

## 3. Results

In total, 309 chars were caught in Badesø during the four sampling years, 144, 77, 54, and 34 chars in 2008, 2013, 2018, and 2019, respectively. In Langesø, 391 chars were caught during the two years of sampling, 258 and 133 chars in 2013 and 2019, respectively. The char CPUE was significantly higher in Langesø than in Badesø (ANOVA, $F_{df = 1.51} = 19.78$, $p < 0.001$). There was no significant interaction between the year and habitat of the catch. The catch of char decreased significantly in Badesø and Langesø during the study period (Figure 3). The stickleback catch fluctuated in Badesø and was lowest in 2013 and highest in 2018 (Figure 3). In Langesø, the stickleback catch was significantly lower in 2013 than in 2019. In Badesø, the char catches were significantly higher in the littoral and profundal habitats than that in the pelagic habitat (Figure 3). The char catches in Langesø did not vary significantly between the habitats (Figure 3). For both Badesø and Langesø, the stickleback catches were highest in the littoral habitat, lower in the profundal habitat, and nonexistent in the pelagic habitat (Figure 3).

The ANOVA tests of the log-transformed fork length with time (sample year) did not reveal statistically significant relationships between the fork length of char and the year in which they were caught for either of the lakes. The length distributions were largely unimodal in both lakes, except for slight peaks in 2013 and 2019 in Badesø, where a smaller second mode was observed among the larger char (Figure 4). More large fish were caught in the profundal and pelagic habitats than in the littoral habitat in Badesø, but there was no difference in the length between the catches in the different zones in Langesø (Figure 5).

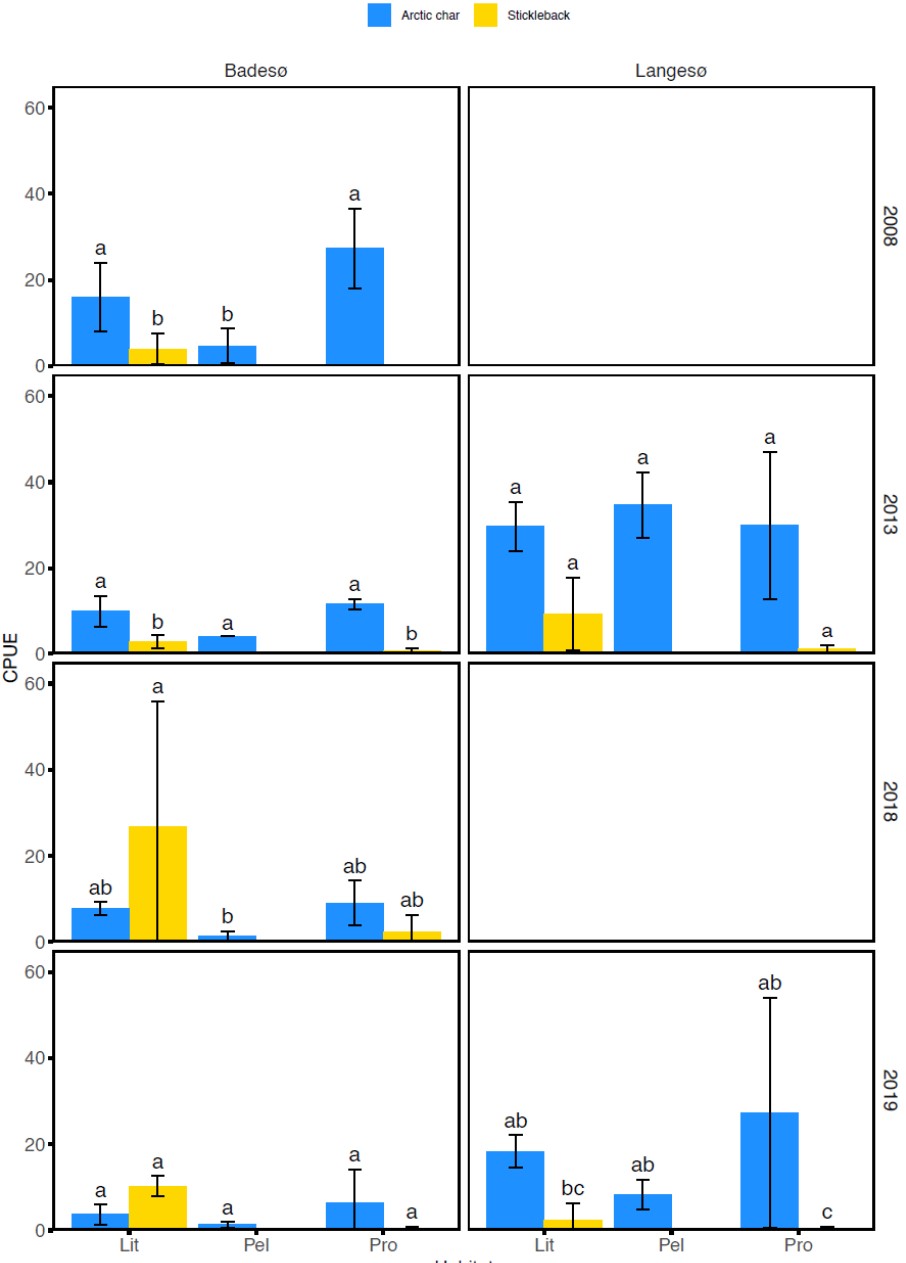

**Figure 3.** Average number of fish caught per net per sampling period, i.e., 16–18 h (CPUE) in the littoral (Lit), pelagic (Pel), and profundal (Pro) habitats in Badesø in 2008, 2013, 2018, and 2019 as well as 2013 and 2019 in Langesø. Error bars show the standard deviation of fish catches between the individual nets in each habitat, as calculated from log-transformed catches. Significant relationships are indicated with letters.

### 3.1. Stomach Content Analysis

In total, 395 Arctic char stomachs (206 from Badesø and 189 from Langesø) were examined during the study. Twelve different item groups (chironomid pupae, chironomid larvae, adult chironomid, zooplankton, three-spined stickleback, Coleoptera, char, Trichoptera, unknown adult Diptera, Hymenoptera, detritus, and sand/gravel) were identified in these stomachs.

In Badesø, a significant variation in stomach content was observed for both large and small chars (ANOSIM, R = 0.727, $p$ = 0.001). Two-way crossed ANOSIM tests were performed for Badesø, where each of the factors "Size", "Habitat", and "Year" were analyzed against each other. For the small chars in Badesø, the analyses showed significant

differences between 2008–2013 and 2018–2019 (ANOSIM, R = 0.763, *p* = 0.01), while no significant difference appeared between "Years" or "Habitats" for the large chars.

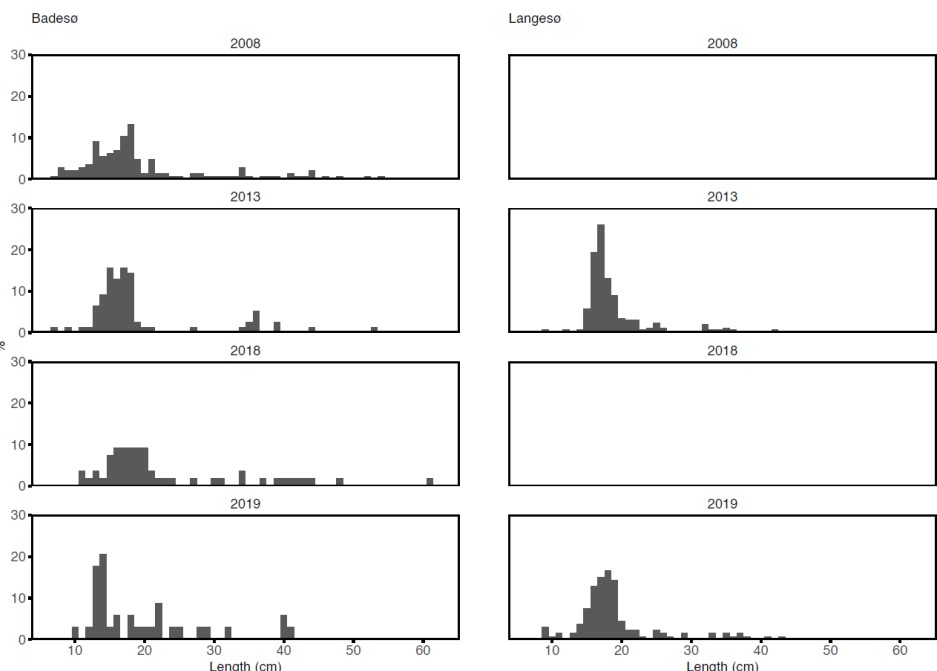

**Figure 4.** Frequency distribution of Arctic char fork lengths in Badesø (2008, 2013, 2018, and 2019) and Langesø (2013 and 2019).

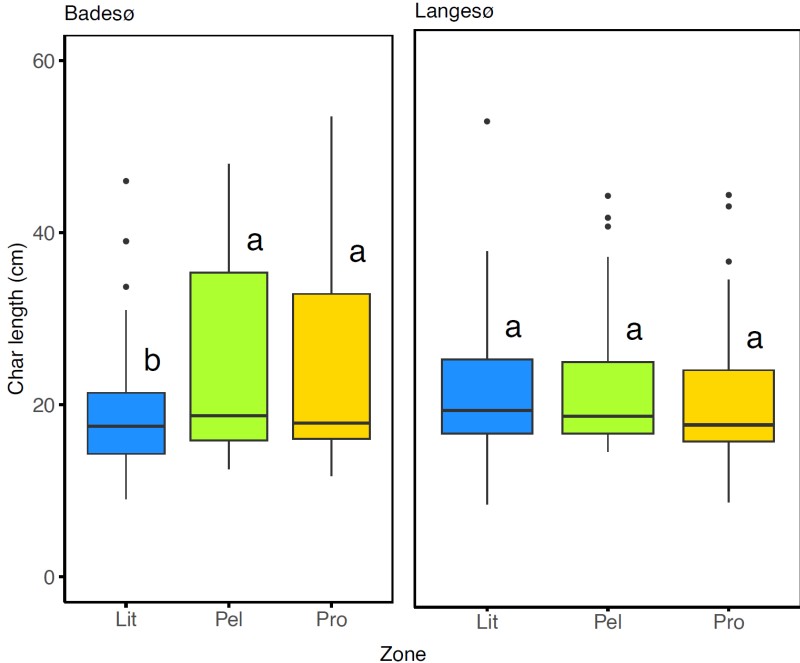

**Figure 5.** Boxplot of Arctic char fork length (cm) per habitat (Lit: littoral, Pel: pelagic, and Pro: profundal) in Badesø and Langesø. Data from all survey years are combined. Different letters indicate significant differences in char length. Lines represents minimum and maximum quartiles, while the box ends represent upper and lower quartiles, and the line median values. Points indicate outliers.

In Badesø, the large chars clustered together based on their dependence on three-spined sticklebacks as a food resource (Figure 6). Only the small chars from the pelagic habitat in 2008 broke this pattern and shared general similarities in their diet with the large

chars (piscivory). This group comprised only four individuals, one of these being a small piscivorous char (Figure 2). The small chars caught in Badesø in 2008–2013 consumed more detritus and zoobenthos (chironomid larvae, adults, and pupae) and clustered together, while the small chars from 2018–2019 consumed more zooplankton and formed a separate cluster (Figure 6). The PCA (Figure 6) confirmed the influence of fish length on this diet. The small chars consumed mostly zoobenthos and/or zooplankton and formed two distinct clusters relative to year (Figure 6). However, the PCA analysis of the char stomach content also revealed two trophic niches that were separated by the exploitation of three-spined sticklebacks as a food resource (Figure 6).

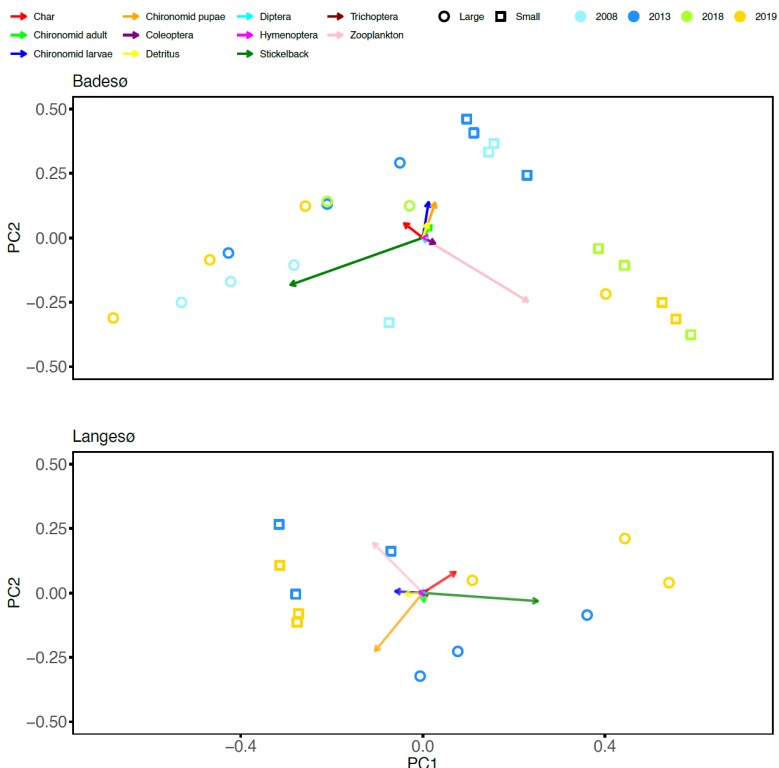

**Figure 6.** Principal component analysis (PCA) scores plot of the stomach content of Arctic char in Badesø (upper) and Langesø (lower). The two PCAs were based on char fork length (<20 cm and >20 cm), time (between years), and habitat. Habitat was not significant and thus not included in the figures. The arrows show how the individual diet items influenced the principal components.

The small chars in Langesø primarily had zoobenthos in their diet. Furthermore, there was significant difference between the stomach content of the large and small chars in Langesø (ANOSIM, R = 0.922, *p* = 0.002). The two-way crossed ANOSIM tests showed a significant difference in the diets of large chars in Langesø between 2013 and 2019 (ANOSIM, R = 0.519, *p* = 0.03). The large chars consumed more zoobenthos in 2013, while they were more piscivorous in 2019 (Figure 6).

### 3.2. Isotopes and Simmr Mixing Model

Linear regression models for $\delta^{15}N$ and $\delta^{13}C$, with the factors of log-length and time (between years), were performed for both lakes. For $\delta^{13}C$ in Badesø, length had the highest positive effect on the correlation, while the year 2008 had a significant negative effect (Table 2).

For $\delta^{15}N$ in Badesø, length had the highest positive effect on the correlation in all the years (Table 2). For Langesø, length had the highest positive effect on the correlation of $\delta^{15}N$, while year did not have a significant effect (Table 2). For $\delta^{13}C$, length had the highest positive effect, but year also had a significant positive correlation (Table 2).

**Table 2.** Linear regressions for Badesø and Langesø with the response variables $\delta^{13}$C and $\delta^{15}$N and the factors of fork length (log-transformed) and time (between years).

|  |  | Variables | Coefficients | Std. Error | *t*-Value | *p* |
|---|---|---|---|---|---|---|
| Badesø | $\delta^{13}$C | Intercept (2008) | −31.3 | 0.78 | −40 | <0.001 |
|  |  | Fork-length | 5.7 | 0.58 | 9.9 | <0.001 |
|  |  | 2013 | 0.18 | 0.27 | 0.67 | 0.507 |
|  |  | 2018 | −1.7 | 0.3 | −6.3 | <0.001 |
|  |  | 2019 | −0.3 | 0.31 | −0.97 | 0.333 |
|  | $\delta^{15}$N | Intercept (2008) | −2.7 | 0.33 | −8.09 | <0.001 |
|  |  | Fork-length | 7.42 | 0.24 | 30.7 | <0.001 |
|  |  | 2013 | 0.91 | 0.11 | 8.04 | <0.001 |
|  |  | 2018 | 0.45 | 0.11 | 3.97 | <0.001 |
|  |  | 2019 | 0.79 | 0.13 | 6.07 | <0.001 |
| Langesø | $\delta^{13}$C | Intercept (2013) | −29.2 | 1.45 | −20.1 | 0.001 |
|  |  | Fork-length | 3.6 | 1.07 | 3.34 | <0.001 |
|  |  | 2019 | 0.68 | 0.32 | 2.11 | 0.036 |
|  | $\delta^{15}$N | Intercept (2013) | −0.17 | 0.77 | −0.22 | 0.112 |
|  |  | Fork-length | 6.34 | 0.57 | 11.19 | <0.001 |
|  |  | 2019 | 0.27 | 0.17 | 1.58 | 0.117 |

The three-spined sticklebacks caught in gillnets from the littoral habitat had significantly higher $\delta^{15}$N values than the small chars (ANOVA, $F_{df = 1.201}$ = 59.91, $p < 0.001$). The small sticklebacks (<2 cm total length) were caught extraordinarily using a scoop net in Badesø in 2008 and Langesø in 2019, and these individuals had a significantly lower $\delta^{15}$N than the larger gillnetted sticklebacks (>4 cm in total length; ANOVA, $F_{df = 1.109}$ = 103.4, $p = 0.001$).

The char data used in the mixing model are shown in Figure 7. All the char isotope values fell within the range expected based on the observed prey isotope values and the model achieved a good convergence.

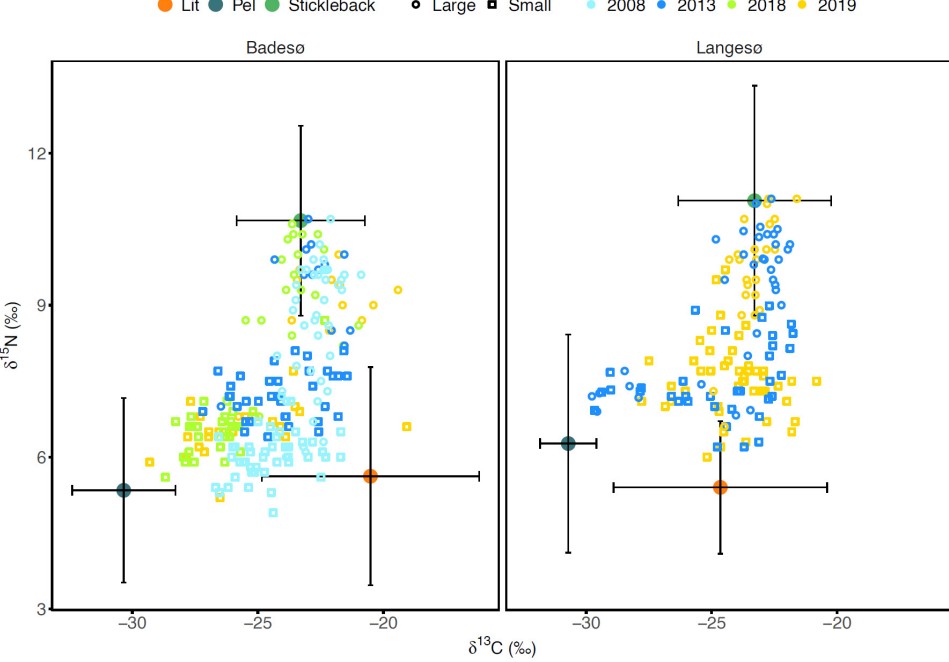

**Figure 7.** Isospace plots of diet contents of large and small Arctic chars based on simmr mixing model analyses. The data show the isotopic values of the sources, corrected by a trophic enrichment factor: littoral invertebrates (lit), pelagic zooplankton (pel), and sticklebacks. Error bars on sources are standard deviation. All char values fell within the isotopic range of the chosen prey sources.

The relative contributions of the different food sources to the diets of the Arctic chars are shown in the ternary plots in Figure 8. The large chars were consistently piscivorous in all the years in Badesø, but less so in 2008 than in subsequent years. The small chars in Badesø had a mixed diet of littoral invertebrates and pelagic zooplankton in the earlier study years of 2008 and 2013. In the later years, 2018 and 2019, pelagic zooplankton was the most important diet source for the small chars in Badesø. Like in Badesø, the large chars in Langesø were mainly piscivorous. The small chars in Langesø relied more heavily on littoral invertebrates in their diet than those in Badesø. There was no difference in the stomach contents between the two investigated years in Langesø.

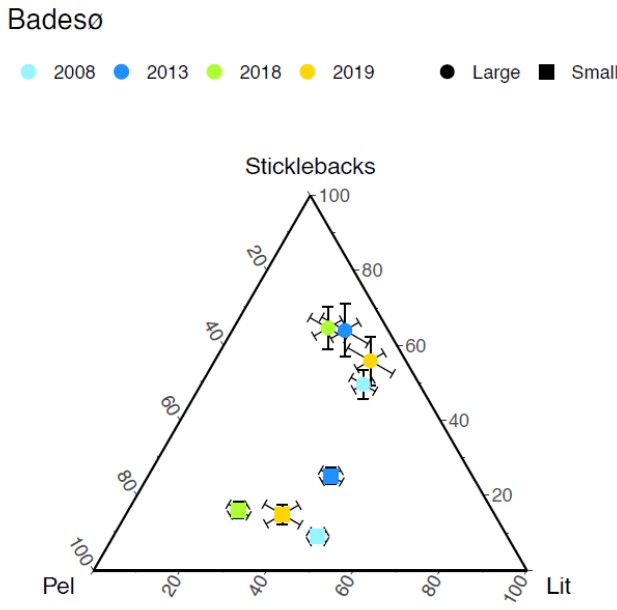

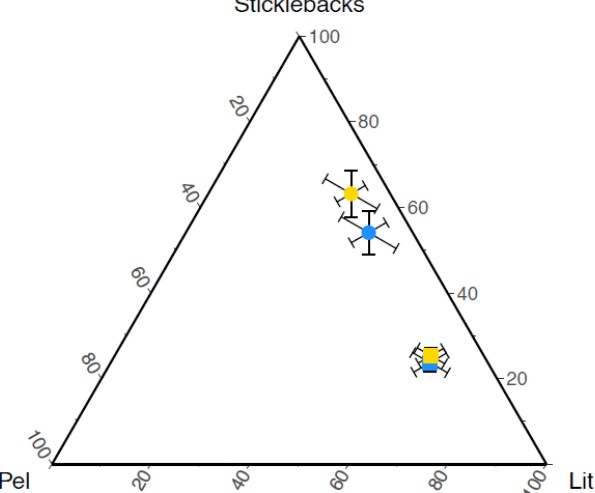

**Figure 8.** Ternary plots of the contribution (%) of piscivory, as well as littoral and pelagic invertebrates, as determined by a simmr mixing model for large and small Arctic chars in Langesø and Badesø.

## 4. Discussion

As is also the case for fish populations in temperate climate zones, the population structure of the Arctic char in Greenland has been found to be influenced by lake morphometry [15]. The char CPUE was significantly lower in the pelagic habitat of Badesø; this was

not observed in the smaller Langesø. This strongly suggests that the fish density is lower in the open waters of larger, deeper lakes. This could be caused by a higher food availability in the more productive littoral environment [27], while in smaller and more elongated lakes, the habitats are so closely connected that fish move freely between them. The char catches decreased over the study period in both lakes. We deemed that this decline was likely not caused by environmental factors, based on the gradual nature of the decline, which did not correspond to similar changes in the environment.

Lakes deeper than 20 m often have bimodal fish length distributions, indicating well-defined populations of large chars. Accordingly, we expected to find a well-defined bimodal fish length distribution in the relatively large and deep Badesø (max depth 38 m) and a unimodal fish length distribution in the comparatively smaller and shallower Langesø (max depth 16 m). However, the length distributions were unimodal and similar in both lakes, except in 2013, where a small mode of large char was found in Badesø. Bimodal char populations are found in Arctic lakes with only char and a size bottleneck develops due to cannibalism [15,16]. However, three-spined sticklebacks were present in both our study lakes, likely alleviating the predation on small chars.

We found that the most significant differences in the stomach content were those between the large and small chars, providing clear evidence of an ontogenetic diet shift towards piscivory in larger fish. This shift is consistent with the findings of other studies on char populations [16,17,19,44].

The stomach content analysis did not reveal significant diet differences between the habitats within the two lakes. This was likely because of the high similarity between the prey items at the resolution used in the study of the different zones (to some degree, chironomid larvae, zooplankton, and sticklebacks all occurred in two or more habitats). However, the chars generally had large proportions of zoobenthos in their diet, implying extensive feeding in the littoral zone, as has also been seen in studies on similar lake types, thus demonstrating the importance of the littoral habitat [22,29,45].

The lengths of the chars caught in the littoral zone of Badesø were significantly smaller than those in the profundal and pelagic zones. This provides evidence for a habitat shift based on life stage, which is in accordance with Klemetsen et al. (1989) [46] and is most likely owed to the fact that predation is lower closer to the substrate, as found by L'Abée-Lund et al. (1993) [47], and that Arctic chars feed heavily in the more profitable littoral habitat [12,47].

In Badesø, the stomach content analysis showed two distinct clusters of small chars, depending on the sampling year. In 2008 and 2013, small chars consumed remarkably more zoobenthos than those in 2018 and 2019, implying a shift towards a more zooplankton-rich diet and, consequently, pelagic feeding in recent years. Whether this shift towards pelagic feeding reflects an increase in pelagic production is an open question. Large-bodied zooplankton have not shown obvious changes in their population dynamics (unpublished data), but this does not mean that pelagic production did not increase, as the predation pressure on zooplankton is generally high in clear, nutrient-poor Greenlandic lakes with the presence of char and three-spined sticklebacks [8]. In Badesø, the analyses revealed no statistical difference in the diets of the large chars over time, as they relied heavily on sticklebacks as their primary food source in all years.

In Langesø, the diets of the small chars did not change between 2013 and 2019, with their main food items being zoobenthos and zooplankton. However, differences appeared in the diets of the large chars. In 2013, the large chars mainly consumed chironomid pupae and sticklebacks, while in 2019, they mostly consumed sticklebacks and chars. The most common lifespan of adult chironomids is 1–2 weeks [48] and their abundance in the Arctic is highest from mid to late July [49,50]. This could explain why chironomid larvae were a more optimal/available food source for the large chars in 2013 than in 2019, as the sampling in 2013 was undertaken in early August, but in mid-July in 2019, the latter potentially coinciding with a high emergence in the local area. This also explains why the high proportion of chironomid pupae in these diets was not reflected in the stable isotope

analysis. Furthermore, there was a high degree of cannibalism by the large chars in Langesø in 2019, which is probably a density-determined behavior, as well as due to the possibly low availability of other prey items and/or competition for these [51].

The stomach content analyses demonstrated important differences in the diet structures between the two lakes, likely reflecting the morphometric differences. The littoral zone is the most important habitat for the foraging of predatory fish in the Arctic [29,52]. The relative size of the littoral zone is larger in Langesø than in Badesø and the ratio of zoobenthos to zooplankton is therefore presumably higher. This might be the reason for the absence of a switch to higher zooplanktivory in Langesø. The morphometry and size of Badesø made the pelagic habitat more important as a foraging area for the chars. Accordingly, zooplankton were the most important food resource for small chars, constituting more than 50% of their stomach contents in 2018 and 2019. This corresponds to the results of Eloranta et al. (2013) [12] and shows that zooplankton had an important role, at least periodically, in subsidizing the diets of primarily benthivorous chars.

There was a positive relationship between the Arctic char length and $\delta^{15}N$ in both lakes, demonstrating that chars forage at increasingly higher trophic levels with increasing lengths.

Three-spined sticklebacks generally had a higher $\delta^{15}N$ than small chars, despite the fact that the small chars were several times larger than the sticklebacks. A higher $\delta^{15}N$ reflects resource partitioning between the sticklebacks and small Arctic chars, as also found in a study by Jørgensen & Klemetsen (1995) [53]. Among the sticklebacks, the small individuals (<2 cm) caught in Badesø in 2008 had significantly lower $\delta^{15}N$ values than the larger net-caught sticklebacks and a higher $\delta^{13}C$ than small chars. This may reflect that large sticklebacks prefer copepods with a higher $\delta^{15}N$ than other zooplankton [8]. It is likely that three-spined sticklebacks also undergo an ontogenetic shift in their lifecycle, as observed in a similar system with nine-spined sticklebacks [29].

The littoral zone has been found to be the optimal foraging habitat for chars [29,31,46,47]. The littoral habitat was the main foraging habitat for the majority of the chars in Langesø, as indicated by the simmr mixing model, but was less important in Badesø. We strongly believe that these differences stem from the morphometric differences between the lakes.

Our results suggest that the littoral habitat is likely the most important habitat for Arctic chars, as suggested by Klemetsen et al. (1989), L'Abée-Lund et al. (1993), Karlsson & Byström (2005), Eloranta et al. (2010), and Eloranta et al. (2013) [12,29,31,46,47]. However, we found zooplankton to be of a high importance for the foraging of small chars in the lake with an increased proportional pelagic area. These results support the results of Eloranta et al. (2015) [19], who found the littoral reliance of Arctic char in Finnish lakes to be negatively correlated with increasing lake size.

The results of the stomach content and stable isotope analyses corresponded well and showed that the large chars in both lakes were piscivorous, but also consumed zoobenthos. The stomach content analysis revealed that the small chars in Badesø shifted from eating mainly zoobenthos in 2008 and 2013 to eating mostly zooplankton in 2018 and 2019. In accordance with this, the simmr mixing model found that the small chars in Badesø were more planktivorous in 2018 and 2019 compared to 2008 and 2013, indicating more extensive open-water feeding, supporting the results from the stomach content analysis. Together, the stomach content and stable isotope analyses provide strong evidence for a dietary niche shift of the small chars in Badesø between 2013 and 2018. This could be caused by an increase in the pelagic zooplankton abundance in the late open-water season [54–56]. However, this is unlikely, based on that the fact that this change towards zoobenthivory was also observed in the stable isotope results. The slow turnover of fish muscle tissue would suggest that the change towards zoobenthivory is symptomatic of more than just a short-term change.

While the stomach content analysis of the large chars in Langesø showed that they relied significantly more on zoobenthos in 2013, it revealed a higher degree of piscivory in 2019. The isotope data, however, showed no statistical difference, even though the proportion of piscivory found by the simmr model was slightly higher. This may reflect

that the sampling in 2019 was undertaken when chironomids had emerged, highlighting the difficulty of observing short-term changes in a stable isotope analysis. Changes in the isotope values from a diet that changes only periodically might be difficult to detect, as the assimilation of new isotope ratios in tissue can take time. This emphasizes that an isotope analysis can miss short-term opportunistic changes in diets and strengthens the argument of Persson & Hansson (1999) [30], that a stable isotope analysis should be used in combination with other methods and with caution.

## 5. Conclusions

Our results showed a significantly higher proportion of pelagic feeding in the small chars in the larger and deeper Badesø, while the chars in the smaller and narrower Langesø had significantly higher feeding rates for littoral macroinvertebrates. Furthermore, the small chars caught in the littoral zone of Badesø were significantly smaller than the chars caught in the profundal and pelagic zones, indicating that the small chars in Badesø preferred the littoral zone to the pelagic and profundal zones. These findings highlight how relatively small differences in lake size and depth can influence the importance of habitats for fish foraging. By contrast, the larger fish in both lakes were primarily piscivorous, regardless of the differences in the morphologies of the lakes.

Among the small-sized chars in Badesø, there was a shift from the consumption of mainly zoobenthos to primarily zooplankton in the later years of the study. This indicates a shift towards pelagic reliance, potentially entailing a shift in the food web structure of the lake.

**Author Contributions:** Conceptualization, A.S.B., T.L.L. and K.R.; methodology, A.S.B., T.L.L. and K.R.; validation, A.S.B., K.R. and T.L.L.; formal analysis, A.S.B. and P.G.; investigation, A.S.B. and P.G.; resources, T.L.L., K.R. and E.J.; data curation, A.S.B.; writing—original draft preparation, A.S.B.; writing—review and editing, A.S.B., K.R., P.G., T.L.L. and E.J.; visualization, A.S.B.; supervision, T.L.L.; project administration, T.L.L. All authors have read and agreed to the published version of the manuscript.

**Funding:** Collection of data was funded by the Greenland Ecosystem Monitoring program (GEM) and the study relied on those data. ASB was funded through a Sino-Danish Centre for Education and Research PhD fellowship, KR and TLL were funded through the GEM program, EJ was funded through the TÜBITAK outstanding researcher program BIDEB2232 (project 118C250) and the Carlsberg Foundation (CF18-0894).

**Data Availability Statement:** Data are available from the authors upon reasonable request.

**Acknowledgments:** We would like to thank Mikko Kiljunen for additional guidance on stable isotope analysis.

**Conflicts of Interest:** The authors declare no conflict of interest. The funders had no role in the design of the study; in the collection, analyses, or interpretation of data; in the writing of the manuscript; or in the decision to publish the results.

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
