# Peer review of "The Importance of Habitat and Lake Morphometry for the Summer Diet Choice of Landlocked Arctic Char in Two West Greenland Lakes"

_water, doi:10.3390/w15122164_

Round 1
Reviewer 1 Report
The overall quality of the manuscript is good. The presented results are concise. I suggest a few modifications which could help to think more about the results and improve the clarity of presentation and data analyses.
The manuscript authors present data on arctic char feeding preferences in Greenland lakes. Diet observations continued from 2008 to 2019, and then the monitoring program ceased. Or at least, no information is provided about the continuation of monitoring outside of this period. If long-term change due to climate warming is anticipated, and therefore the monitoring is performed, that would be interesting information to place the paper results into context. Presumably, similar monitoring exists, and a hypothesis could be built about anticipated changes in the Arctic food web. Also the integration of SIA and stomach content analyses result could be elaborated.
1. Line 91. The trophic position is a key element of the presented research. It could be calculated based on δ15N data and support the conclusion. Why it is not done so? In Lines 247-248 it is stated that ‘two trophic levels that were separated by the exploitation of three-spined sticklebacks as a food resource’ were distinguished. Do you mean TL 3 and TL4? How this is reflected in SIA data and what is the TL i.e. the position in the food chain of the char? According to mixing models, the piscivory level is the same regardless that the difference between the ultimate use of carbon from the littoral vs. pelagic source is significant between two lakes. I guess, the share of fish in the diet does not necessarily reflect trophic position in the food chain, I suggest providing TL values in this study. And plot TL values with BL in two different lakes.
2. Line 287. ‘Trophically enriched sources’ change to ‘The isotopic values of the sources are corrected by a trophic enrichment factor’.
3. In Figure 6 I wonder if PCA plot of the lake Langesø could be rotated so that PC1 in bot graphs could correspond to increasing predation i.e. share of Stickleback in the diet?
4. Lines 263-267. This paragraph is not clear. How separate years could be used as factors? If time is a factor then years are levels of the factor. If time as a factor is significant (and there is long-term population change behind), then I would recommend proceeding with post-analyses and look which year exactly is different. It is confusing and difficult to understand what ecological meaning is behind year (not environmental variable) having negative or positive effects.
5. Integration of SIA and diet analysis results and formulation could be improved. For example, Line 254 Statement ‘Large char consumed more zoobenthos in 2013, while they were more piscivorous in 2019 (Figure 6)’. While Lines 270-272 statement ‘Langesø length had the highest positive effect on the correlation of δ15N, while year did not have a significant effect (Table 2).’ If there is a contradiction between two methodologies, why it is so?
6. Line 266. The year 2008 should be year 2018. Or year 2008 is not included in Table 2?
7. Figure 7, Lines 282-284. Why char is not included as a source? Is it assumed that stickleback SI values are equal to char? If so pooling of stickleback and small char values should be performed. Otherwise, the information in Figure 8 is inconsistent with Figure 7.
8. Littoral invertebrates have more enriched δ13C values in Badesø than Langesø, why it could be so? What implication does this differentiation between lake littoral vs. pelagic source have to SIA methodology application?
9. Line 329. Double ‚In‘ should be removed.
10. Repetitive information Lines 363-365: ‚The littoral zone is the most important habitat for the foraging of predatory fish in the Arctic [29, 52]. The relative size of the littoral zone is larger in Langesø than in Badesø and the ratio of zoobenthos to zooplankton therefore presumably higher.‘ And Lines 389-391: ‚Production in Arctic lakes is often highest in the littoral habitat [27]. The relative size of the littoral zone is larger in Langesø than in Badesø and the ratio of zoobenthos to zooplankton therefore presumably higher in Langesø.
Author Response
Thank you for taking the time to review our paper, please find our responses to your comments in the attached file.

Reviewer 2 Report
Dear Authors,
As the current study aims to explain the effects of habitat (littoral, pelagic, or deep) and lake morphometry on the diet and trophic position of Arctic char (Salvelinus alpinus) in two Greenlandic lakes, Badesø and Langesø, during the summer season; I consider the objectives to be achieved.
This research clarifies the changes in the nutritional spectrum as ontogenetic diet shifts of Arctic char and reflects the changes at the population level in both lakes. The changes in the CPUE of the fish in the two lakes are well explained, as well as shifts in the zoobenthos/zooplankton ratio in food, and the level of predation depending on the characteristics of the habitat and its morphometry have been clarified.
I have only one remark:
In the Discussion section:
Line 329, delete one in from “…..used in in the study….. ”
Author Response

(The authors gave the same response as above.)

Round 2
Reviewer 1 Report
The isotopic niche could be presented and calculated using a standard ellipse area (Jackson et al., 2011, 2012).
I'm not fully convinced about changing the tropic level to a trophic niche term. But I leave it to decide by the editor.